# Preferential Expansion of HPV16 E1-Specific T Cells from Healthy Donors’ PBMCs after Ex Vivo Immunization with an E1E2E6E7 Fusion Antigen

**DOI:** 10.3390/cancers15245863

**Published:** 2023-12-15

**Authors:** Joana Daradoumis, Mikkel Dons Müller, Patrick Neckermann, Benedikt Asbach, Silke Schrödel, Christian Thirion, Ralf Wagner, Per thor Straten, Peter Johannes Holst, Ditte Boilesen

**Affiliations:** 1InProTher ApS, Bioinnovation Institute, Ole Maaløes Vej 3, 2200 Copenhagen, Denmark; mikkel@sund.ku.dk (M.D.M.); pholst@hervolutiontx.com (P.J.H.); 2Department of Immunology and Microbiology, University of Copenhagen, Blegdamsvej 3B, 2200 Copenhagen, Denmark; 3Institute of Medical Microbiology & Hygiene, Molecular Microbiology (Virology), University of Regensburg, Universitätsstraße 31, 93053 Regensburg, Germany; 4Sirion Biotech GmbH, 82166 Gräfelfing, Germany; 5Institute of Clinical Microbiology and Hygiene, University Hospital Regensburg, Franz-Josef-Strauß-Allee 11, 93053 Regensburg, Germany; 6Center for Cancer Immune Therapy, Department of Oncology, Copenhagen University Hospital, 2730 Copenhagen, Denmark; 7Loma Therapeutics ApS, Bioinnovation Institute, Ole Maaløes Vej 3, 2200 Copenhagen, Denmark

**Keywords:** human papillomavirus, adenoviral vectors, therapeutic vaccines, E1, T-cell expansion, dendritic cells

## Abstract

**Simple Summary:**

The translational gap poses a challenge for therapeutic cancer vaccine development, including HPV-specific therapeutic vaccines. The aim of this study was to evaluate a new human ex vivo PBMC assay that mimics how viral-vectored vaccines induce T-cell responses in vivo. In brief, PBMCs are taken through rounds of stimulation by co-culturing with syngeneic DCs, which have been previously matured and transduced with the viral-vector-based vaccine. We evaluated the assay using a novel HPV16-specific therapeutic vaccine in healthy human donors with pre-existing T-cell responses against HPV16 to simulate a therapeutic setting. The pre-existing T-cell responses were effectively boosted to a larger extent by vaccine-transduced DCs than by peptide-pulsing, and the T cells could specifically recognize and kill HPV16^+^ human cancer cells. Additionally, this study supports the use of nucleic-acid-based vaccines for optimal cancer recognition and indicates an immunological advantage of targeting the HPV E1 gene.

**Abstract:**

Persistent human papillomavirus (HPV) infection is responsible for practically all cervical and a high proportion of anogenital and oropharyngeal cancers. Therapeutic HPV vaccines in clinical development show great promise in improving outcomes for patients who mount an anti-HPV T-cell response; however, far from all patients elicit a sufficient immunological response. This demonstrates a translational gap between animal models and human patients. Here, we investigated the potential of a new assay consisting of co-culturing vaccine-transduced dendritic cells (DCs) with syngeneic, healthy, human peripheral blood mononuclear cells (PBMCs) to mimic a human in vivo immunization. This new promising human ex vivo PBMC assay was evaluated using an innovative therapeutic adenovirus (Adv)-based HPV vaccine encoding the E1, E2, E6, and E7 HPV16 genes. This new method allowed us to show that vaccine-transduced DCs yielded functional effector T cells and unveiled information on immunohierarchy, showing E1-specific T-cell immunodominance over time. We suggest that this assay can be a valuable translational tool to complement the known animal models, not only for HPV therapeutic vaccines, and supports the use of E1 as an immunotherapeutic target. Nevertheless, the findings reported here need to be validated in a larger number of donors and preferably in patient samples.

## 1. Introduction

Human papillomavirus (HPV) is the causative agent of a range of cancers, including practically all cases of cervical cancer [1] and about 70% of oropharyngeal cancers (oropharyngeal squamous cell carcinoma, OPSCC) [2,3], as well as most vaginal, anal, penile, and vulvar cancers [4]. Among all HPV types, the HPV type 16 (HPV16) high-risk strain is strongly associated with the majority of HPV-related cancers [5], being responsible for approximately 50% of cervical cancers and more than 80% of HPV^+^ head and neck cancers worldwide [6].

HPV-associated cancers represented nearly 800,000 new cancer cases in 2020, with cervical cancer accounting for about 600,000 of these [7]. While prophylactic vaccination of adolescent women and men, together with cervical cancer screening, are strong tools for the prevention of HPV disease, the global burden of HPV disease will persist for decades to come due to low prophylactic vaccine coverage and lack of access to screening [8,9]. In fact, the incidence of OPSCC is increasing, especially in developed countries [10], where, in some, OPSCC-related mortality has surpassed that of cervical cancer [11].

Therapeutic vaccines that induce CD8^+^ T cells capable of specifically killing tumor cells and malignant cells constitute a type of immunotherapy with great potential for cancer treatment. HPV16 constitutes a suitable exogenous cancer-specific target for therapeutic vaccination of the highest medical importance. Consequently, several therapeutic vaccine candidates have been developed to combat HPV-associated diseases, especially advanced cervical and OPSCC and pre-cancerous lesions of the cervix. The vast majority of these therapeutic vaccines have targeted the HPV oncogenes, E6 and E7, and most of them showed some improvements in clinical outcomes, although often only in about 50% of patients [12,13,14]. Interestingly, the clinical studies on HPV therapeutic vaccines show a clear correlation between the induction of strong anti-HPV T-cell responses and improved clinical outcomes. Here, we pinpoint two specific challenges of the current HPV therapeutic vaccine development and propose potential solutions to advance the field and improve the efficacy of therapeutic vaccines against HPV-associated diseases.

First, the available preclinical models constitute a considerable translational challenge [13]. Outbred mice are useful for the assessment of the immunogenicity of therapeutic vaccines, but they cannot be used for efficacy evaluation. The commonly used models to assess therapeutic efficacy are the C57BL/6-based murine transplant models, which are inherently limited in their major histocompatibility complex (MHC)-diversity, in contrast to the highly MHC-diverse human population. As HPV16 E7 harbors a strong C57BL/6 epitope [15], the use of this model is likely to lead to an overestimation of the potential to induce anti-tumor efficacy. *Macaca Fascicularis* with cervical papillomavirus infection also provides a highly relevant model for the treatment of cervical infection [16] but does not address the cancer stages of HPV disease and is also less commonly used due to its availability and ethical concerns.

As a supplement to the commonly used murine efficacy models, we present here a novel human ex vivo T-cell expansion assay, which aims to mimic certain aspects of a human in vivo immunization and allows us to obtain high percentages of antigen-specific T cells to perform further assays, such as investigating the vaccine immunogenicity, epitope hierarchies, and tumor-killing capacity.

Second, there has been little innovation in terms of the antigenic targets of HPV therapeutic vaccines. Generally, a good antigen is defined as being immunogenic but also by not being subject to prior exhaustion by chronic infection and/or cancer [17]. Increasing the amount of antigen included in a therapeutic vaccine may enable more patients to benefit from the treatment by enhancing the possible presence of a suitable epitope(s) across a high variety of human leukocyte antigens (HLA) alleles. We propose a shift away from the narrow focus on E6 and E7 as targets that dominate the field of HPV therapeutic vaccines and suggest targeting E1 and E2 in addition to E6 and E7 and incorporating full-length antigens rather than selected epitopes [13]. We have previously shown that such a vaccine design is capable of inducing strong T-cell responses in outbred mouse strains and have seen indications that E1-specific T cells may enhance tumor control in C57BL/6 mice. The use of E1 and E2 as targets for HPV therapeutic vaccines is also supported by recent studies on patients with cervical cancer or OPSCC [18,19,20].

Here, we show that HPV16-specific T cells can be expanded in HPV16-reactive healthy donors’ peripheral blood mononuclear cells (PBMCs) in an ex vivo model of immunization using dendritic cells (DCs) transduced with an adenoviral-vectored therapeutic vaccine encoding HPV16 E1, E2, E6, and E7 genes [21]. We show that adenoviral-vectored transduction of DCs is superior at expanding T cells compared to the peptide co-culture, and we observe evidence that HPV16 E1-specific CD8^+^ T cells are reliably inducible and dominate over time in non-biased T-cell expansion cultures. This indicates that E1 is the most immunogenic of the four HPV16 antigens and induces T cells of the highest replicative fitness.

In this work, we make use of the human adenoviral-vector serotypes 19a/64 (Ad19) [22,23], 5 (Ad5) [21,24], and five with the fiber from serotype 35 (Ad5f35) [25,26] to circumvent the widespread pre-existing immunity towards Ad5 in the human population [27,28,29,30]. Ad5 is included in the presented work but is only considered as a research tool and not as a candidate for a clinically relevant therapeutic vaccine due to this high level of pre-existing immunity, which would hamper the vaccine efficacy in humans. On the contrary, the other vectors are well suited for clinical use. Ad19a/64 is a rare human serotype of the D subtype where immunity in humans is scarce and of low titer [22,23], whereas the Ad5f35 contains the Ad5 vector backbone but employs a fiber exchange, which in effect provides immune evasion and, thus, avoids unfavorable Ad5 neutralization in humans [25].

## 2. Methods

### 2.1. Antigen Design and Adenoviral Vectors

HPV16 antigens and additional elements of the vaccines were designed as previously described in P. Neckermann et al.’s publication [31]. E1, E2, E6, and E7 of the HPV16 sequences (gi|333031|lcl|HPV16REF.1|HPV16REF) were gene-optimized and synthesized at Geneart Thermo Fisher (Regensburg, Germany) [32]. E1/E3-deficient human adenoviral vectors of serotypes 19a/64 (Ad19) and 5 (Ad5) were generated as previously described [24,31]. Ad5f35 vectors encoded the same antigen as described for the Ad19 and Ad5 vectors and were produced as described in Nilsson et al.’s work [26].

### 2.2. Peripheral Blood Mononuclear Cells (PBMCs) Isolation and Cryopreservation

Buffy coats from anonymous healthy donors were obtained at the blood bank of Rigshospitalet Hospital (Copenhagen, Denmark). Samples were immediately processed for the isolation of peripheral blood mononuclear cells (PBMCs) by centrifugation using the Lymphoprep density gradient medium (07801, STEMCELL, Cambridge, UK) following the manufacturer’s guidelines. Isolated PBMCs were washed three times with 1× PBS by centrifugation (300× *g* 10 min RT). The last washing step (200× *g* 10 min RT) was used to remove platelets. Cells were cryopreserved at −150 °C in the complete RPMI 1640 GlutaMAX medium (61870044, Thermo Fisher Scientific, Roskilde, Denmark) with 20% FBS (F9665-500ML, Sigma-Aldrich, Søborg, Denmark) and 10% dimethyl sulfoxide (DMSO) (D2438, Sigma-Aldrich, Søborg, Denmark) using a CoolCell (CLS432002, Sigma-Aldrich, Søborg, Denmark) gradual freezing device. PBMCs were thawed by gentle agitation in a 37 °C water bath and transferred into 15 mL falcon tubes containing 9 mL of warm R10 medium consisting of RPMI 1640 GlutaMAX medium (61965059, Thermo Fisher Scientific, Roskilde, Denmark) with 10% human serum (H5667, Sigma-Aldrich, Søborg, Denmark), 1 mM sodium pyruvate (11360070, Thermo Fisher Scientific, Roskilde, Denmark), 1% (*v*/*v*) penicillin-streptomycin (15140122, Thermo Fisher Scientific, Roskilde, Denmark), and 10 mM HEPES (15630056, Thermo Fisher Scientific, Roskilde, Denmark). Each falcon tube was subsequently mixed by inversion and centrifuged for 5 min at 300× *g* RT. PBMCs were incubated with 100 µg/mL of DNAse I solution (07900, STEMCELL, Cambridge, UK) at RT for at least 15 min prior to culturing or monocyte isolation.

### 2.3. Monocyte Isolation and Differentiation to Dendritic Cells (DCs)

Monocytes were isolated from the PBMCs using the EasySep Human Monocyte Isolation Kit (19359, STEMCELL, Cambridge, UK) following the manufacturer’s instructions. Briefly, the PBMCs were adjusted to 5 × 10^7^ cells/mL in EasySep Buffer (20144, STEMCELL, Cambridge, UK) and transferred into a 5 mL polystyrene round-bottom tube (38007, STEMCELL, Cambridge, UK). A monocyte isolation cocktail (19359, STEMCELL, Cambridge, UK) was added at 50 µL/mL of the sample and incubated for 5 min at RT. Afterward, the magnetic particles (19359, STEMCELL, Cambridge, UK) were added at 50 µL/mL of the sample and incubated for 5 min at RT. Finally, the samples were topped up to 2.5 mL of EasySep Buffer (20144, STEMCELL, Cambridge, UK), mixed gently and the tubes were placed into the magnet (18103, STEMCELL, Cambridge, UK) for 2.5 min at RT. Cells were carefully transferred into a new tube, which was placed into the magnet (18103, STEMCELL, Cambridge, UK) and incubated for 2.5 min at RT. Cells were carefully harvested and counted using the NucleoCounter NC-202 (ChemoMetec, Lillerød, Denmark). After centrifugation (300× *g* 8 min RT), the isolated monocytes were resuspended in ImmunoCult-ACF Dendritic Cell Medium (10986, STEMCELL, Cambridge, UK) with the 1:100 ImmunoCult™-ACF Dendritic Cell Differentiation Supplement (10988, STEMCELL, Cambridge, UK). Furthermore, 1 × 10^6^ cells/well were cultured in Nunc Multidishes with UpCell™ Surface (174900, Thermo Fiher Scientific, Roskilde, Denmark). On day 3, the ImmunoCult DC Differentiation Medium (10986, STEMCELL, Cambridge, UK) was exchanged, and 2 days later (day 5), DC maturation was induced by adding the ImmunoCult Dendritic Cell Maturation Supplement (10989, STEMCELL, Cambridge, UK) at 1:100.

### 2.4. Transduction of Mature DCs

Meanwhile, 24 h after the addition of the maturation supplement, mature DCs were counted and subsequently transduced with different Adv-vector-based vaccines encoding HPV16 genes with or without Ii: Ad19-Ii-E1E2E6E7, Ad19-E1E2E6E7, Ad5f35-Ii-E1E2E6E7, or Ad19-NegCtrl (empty vector), and were used to assess the background responses, at an MOI of 100. For Ad5-Ii-E1E2E6E7, transduction was completed at an MOI of 5000 in the presence of 1:200 of lentiboost (SIRION Biotech GmbH, Graefelfing, Germany) and 4 µg/mL Sequa-Brene (S2667, Sigma-Aldrich, Søborg, Denmark). Furthermore, 2 h after transduction, the DCs were washed and cultured in ImmunoCult-ACF Dendritic Cell Medium (10986, STEMCELL, Cambridge, UK) without supplements. Matured adenovirus (Adv)-transduced DCs were used for the experiments 24 h after infection. DCs were also harvested 72 h after transduction to assess the expression of the Adv-encoded transgene over time.

### 2.5. Cancer Cell Lines

The epidermoid cervical cancer cell line Ca Ski was purchased from ATCC (CRM-CRL-1550, Teddington Middlesex, UK) and was cultured in complete RPMI 1640 GlutaMAX medium (61870044, Thermo Fisher Scientific, Roskilde, Denmark) supplemented with 10% fetal bovine serum (FBS) (F9665-500ML, Sigma-Aldrich, Søborg, Denmark), 1 mM sodium pyruvate (11360070, Thermo Fisher Scientific, Roskilde, Denmark), 1% (*v*/*v*) penicillin-streptomycin (15140122, Thermo Fisher Scientific, Roskilde, Denmark) and maintained at 37 °C in 5% CO_2_. This cell line was reported to have integrated the HPV16 (and HPV18)-related sequences.

### 2.6. PBMC Stimulation and HPV16-Specific T-Cell Expansion

To screen for HPV16-reactive donors, PBMCs from healthy donors were thawed as described above and seeded at 1 × 10^6^ cells/well in 48-well culture plates in RPMI 1640 GlutaMAX medium (61965059, Thermo Fisher Scientific, Roskilde, Denmark) with 10% human serum (H5667, Sigma-Aldrich, Søborg, Denmark). The PBMCs were primed with 1 µg/mL of HPV16 single peptides (Appendix A) or HPV16 E1 (PM-HPV16-E1, JPT, Berlin, Germany), E2 (PM-HPV16-E2, JPT, Berlin, Germany), E6 (PM-HPV16-E6, JPT, Berlin, Germany), or/and E7 (PM-HPV16-E7, JPT, Berlin, Germany) peptide pools and incubated for 2 h at 37 °C, 5% CO_2_. The excess peptide was washed off with a fresh medium before continuing the cell culture. On day 2, the media were changed to RPMI 1640 GlutaMAX medium (61965059, Thermo Fisher Scientific, Roskilde, Denmark) with 10% human serum (H5667, Sigma-Aldrich, Søborg, Denmark) and 100 IU/mL of IL-2 (130-097-746, Miltenyi Biotec, Lund, Sweden). This was repeated every 2–3 days, and cells were split if needed.

Donors showing responses to any of the HPV16 single peptides or HPV16 E1, E2, E6, or E7 peptide pools were considered HPV16-reactive or HPV16^+^. HPV16-reactive donor 9, donor 13, donor 14, and donor 16 were selected for subsequent T-cell expansion experiments. PBMC-priming stimulation was conducted with either HPV16 HLA-A2-restricted single peptides, HPV16 E1, E2, E6, or/and E7 peptide pools, or with Ad19-Ii-E1E2E6E7-transduced DCs at a 1:10 DC:PBMC ratio. Boost stimulation was conducted 7 or 14 days after the prime with either HPV16 single peptides, HPV16 peptide pools, or with Ad19-Ii-E1E2E6E7-, Ad19-E1E2E6E7-, Ad5-Ii-E1E2E6E7-, or Ad5f35-Ii-E1E2E6E7-transduced DCs at a 1:10 ratio. The cells were cultured in RPMI 1640 GlutaMAX medium (61965059, Thermo Fisher Scientific, Roskilde, Denmark) or X-Vivo 15 (BE02-060Q, Lonza, Vallensbæk Strand, Denmark) with 10% human serum (H5667, Sigma-Aldrich, Søborg, Denmark). A total of 100 IU/mL of IL-2 (130-097-746, Miltenyi Biotec, Lund, Sweden) was added 2 days after stimulation. The media were changed every 2–3 days, and cells were split when necessary.

In line with common terminology for vaccines, “prime” refers to the first vaccine/peptide stimulation, “boost” to all subsequent ones, whereas “stimulation” is used in the context of readout assays (intracellular cytokine staining, ICS).

### 2.7. Sorting of Interferon Gamma-Positive Cells

To isolate and further expand HPV16-specific T cells, PBMCs from donors 13 and 14, primed with Ad19-Ii-E1E2E6E7-transduced DCs and boosted with Ad5- or Ad5f35-Ii-E1E2E6E7-transduced DCs were restimulated with either 1 μg/mL of the HPV16 E1 peptide pool or with Ad5f35-transduced DCs (not matching the previous Adv booster) at a 1:5 ratio of DC:T cells for 5 h at 37 °C in 5% CO_2_. The IFNγ-secreting T cells reacting to HPV16 antigens were sorted using the human IFNγ capture assay kit (130-054-201, Miltenyi Biotec, Lund, Sweden) following the manufacturer’s instructions. Briefly, stimulated cells were collected into 15 mL tubes and labeled with the cytokine catch reagent for 5 min on ice. To allow cytokine secretion, cells were incubated in warm X-Vivo 15 media (BE02-060Q, Lonza, Vallensbæk Strand, Denmark) for 45 min with 5 min interval inversions at 37 °C. Subsequently, the cells were labeled by adding the cytokine detection antibody (PE) (130-054-201, Miltenyi Biotec, Lund, Sweden) for 10 min (on ice), followed by the anti-PE magnetic microbeads (130-054-201, Miltenyi Biotec, Lund, Sweden) for 15 min at 4–8 °C. IFNγ-secreting cells were magnetically separated using MS columns (130-042-201, Miltenyi Biotec, Lund, Sweden) placed into the MACS separator magnet (130-042-109, Miltenyi Biotec, Lund, Sweden). After washing, magnetically labeled cells were flushed directly into T-25 culture flasks for T-cell rapid expansion (REP).

### 2.8. Rapid Expansion Protocol (REP)

To expand the HPV16-specific IFNγ-sorted T cells to high cell numbers, we used the REP based on Holmen Olofsson et al.’s work [33]. Briefly, 2 × 10^7^ feeder cells (a mix of PBMCs from 3 different healthy donors) per T-cell condition were irradiated at 30 Gy and were cultured in 20 mL of X-Vivo 15 media (BE02-060Q, Lonza, Vallensbæk Strand, Denmark) with 5% human serum (H5667, Sigma-Aldrich, Søborg, Denmark), 6000 U/mL of IL-2 (130-097-746, Miltenyi Biotec, Lund, Sweden), 0.6 µg of anti-CD3 (Clone OKT3, 16-0037-85, Thermo Fisher Scientific, Roskilde, Denmark), and 25 mM of HEPES (15630056, Thermo Fisher Scientific, Roskilde, Denmark). Then, the IFNγ-labeled T cells were sorted directly into T-25 culture flasks containing the feeder cell mix. Culture flasks were placed standing in the incubator at 37 °C in 5% CO_2_ for 15 days. Three times per week, half of the media was changed to new X-Vivo 15 media (BE02-060Q, Lonza, Vallensbæk Strand, Denmark) with 5% human serum (H5667, Sigma-Aldrich, Søborg, Denmark) and fresh IL-2 (3000 U/mL) (130-097-746, Miltenyi Biotec, Lund, Sweden). Cells were split if necessary. At last, the cells were frozen, and some were subsequently thawed to assess the specificity of the T-cell expansion.

### 2.9. Surface and Intracellular Cytokine Staining (ICS) and Flow Cytometry

Adv-HPV16-transduced DCs were intracellularly stained to assess the fraction and duration of expression of the Adv-encoded HPV16 antigen. Further, 24 and 72 h after transduction, the DCs were harvested and blocked with the human TruStain FcX (422302, BioLegend, Amsterdam, The Netherlands) diluted to 1:10 in FACS buffer (1% BSA, 0.1% NaN_3_, and PBS) for 15 min at RT. Dead cells were discriminated by adding fixable viability dye eFluor780 (65-0865-14, Thermo Fisher Scientific, Roskilde, Denmark) diluted to 1:1000 in PBS for 30 min at 4 °C. The fixation/permeabilization kit (554714, BD Biosciences, Lyngby, Denmark) was used to permeabilize and fixate the cells for intracellular staining with anti-HPV16 E7 (PE, sc-65711, Santa Cruz Biotechnology, Heidelberg, Germany) at a 1:50 dilution for 30 min at 4 °C. HPV16 E7 antibody was used to show the expression of the whole vaccine-encoded HPV16 antigen upon DC transduction, since all HPV16 E1, E2, E6, and E7 genes are fused together.

The HPV16-antigen-specific T-cell responses and specific recognition of Ca Ski cancer cells after the T-cell expansion assays were performed were assessed by surface and intracellular cytokine staining (ICS). HPV16-primed-only or prime-boosted T cells were restimulated with 1 μg/mL of either HLA-A2-restricted HPV16 peptides (Appendix A), HPV16 E1, E2, E6, or E7 peptide pools, Adv-HPV16-transduced DCs, or Ca Ski cancer cells at a 1:5 DC/Ca Ski:T-cell ratio. Immediately after stimulation, a Golgi plug/protein transport inhibitor containing brefeldin A (555029, BD Biosciences, Lyngby, Denmark) was added at a 1:1000 dilution in X-Vivo 15 media (BE02-060Q, Lonza, Vallensbæk Strand, Denmark). Optionally, an anti-CD107a (PE, Clone H4A3, 555801, BD Biosciences, Lyngby, Denmark) extracellular marker was added at 1:100 dilution. Cells were incubated for 4–5 h at 37 °C. Stimulated cells were stained with cell-surface marker antibodies against CD3 (FITC, Clone HIT3a, 555339, BD Biosciences, Lyngby, Denmark), CD4 (PerCP/Cy5.5, Clone RPA-T4, 560650, BD Biosciences, Lyngby, Denmark), and CD8 (BV421, Clone RPA-T8, 562428, BD Biosciences, Lyngby, Denmark) at a 1:50 dilution in PBS. Fixable viability dye eFluor780 (65-0865-14, Thermo Fisher Scientific, Roskilde, Denmark was added to discriminate dead cells at a 1:1000 dilution. Cells were incubated for 30 min at 4 °C in the dark. Subsequently, the cells were fixed with the fixation/permeabilization kit (554714, BD Biosciences, Lyngby, Denmark) and stained intracellularly with antibodies against IFNγ (APC, Clone B27, 506510, BioLegend, Amsterdam, The Netherlands) and TNFα (PE/Cy7, Clone MAb11, 557647, BD Biosciences, Lyngby, Denmark) at 1:50 for 30 min at 4 °C in the dark.

Flow data were collected on a Fortessa 3 or 5 instrument (BD Biosciences, Lyngby, Denmark) and analyzed using FlowJo software v10.7.1 (Tree Star, Ashland, OR, USA).

### 2.10. Chromium-51 (^51^Cr)-Release Assay

To measure HPV16-specific T-cell-mediated cytotoxicity, we performed a conventional chromium-51 (^51^Cr)-release killing assay. Some of the Ca Ski cancer target cells were incubated with an HLA-A2 blocking antibody (Clone BB7.2, 343302, BioLegend, Amsterdam, The Netherlands) at 10 µg/mL. Subsequently, all Ca Ski cells were labeled with 20 µL ^51^Cr, corresponding to 100 µCi of ^51^Cr (NEZ030S001MC, PerkinElmer, Skovlunde, Denmark) in 100 µL of R10 medium consisting of RPMI 1640 GlutaMAX medium (61965059, Thermo Fisher Scientific, Roskilde, Denmark) with 10% human serum (H5667, Sigma-Aldrich, Søborg, Denmark), 1 mM sodium pyruvate (11360070, Thermo Fisher Scientific, Roskilde, Denmark), 1% (*v*/*v*) penicillin-streptomycin (15140122, Thermo Fisher Scientific, Roskilde, Denmark), and 10 mM HEPES (15630056, Thermo Fisher Scientific, Roskilde, Denmark) for 1 h at 37 °C. Subsequently, the different HPV16 effector T cells were co-cultured with the ^51^Cr-labeled Ca Ski target cells at a 60:1 effector:target (EC:TC) ratio following a 5-time 1:3 dilution for 4 h at 37 °C. After incubation, the amount of radioactivity in the supernatant of the co-culture was measured using a gamma counter (2470 WIZARD2 automatic gamma counter, PerkinElmer, Skovlunde, Denmark). The effector cells consisted of Ad19-Ii-E1E2E6E7, Ad5f35-Ii-E1E2E6E7 prime-boost T cells, stimulated and sorted using either the E1 HPV16 peptide pool or Ad5f35-Ii-E1E2E6E7-transduced DCs subjected to REP for 15 days. The T cells primed and boosted with Ad19- and Ad5-NegCtrl-transduced DCs, as well as freshly isolated PBMCs of the matching donor, were used as negative controls.

## 3. Results

### 3.1. Screening for HPV16-Responding Donors

As therapeutic vaccines aim to treat an existing disease, they must act on top of any naturally induced immune responses. Therefore, we used PBMCs from blood donors with pre-existing HPV16-specific T-cell responses. For their selection, we tested 17 HLA-A2^+^ healthy donors for signs of T-cell reactivity against 11 single peptides corresponding to the previously identified HLA-A2-restricted HPV16 epitopes (Appendix A) and four peptide pools (comprising 15-mers with 11 aa overlap) covering the E1, E2, E6, or E7 proteins of HPV16.

For the screening, PBMCs from each donor were primed by 2 h peptide-pulsing in the presence of low-dose IL-2 (100 U/mL). After 7 days, the PBMCs were restimulated with cognate peptides or pools, and the CD8^+^ and CD4^+^ T-cell responses were measured by ICS and flow cytometry. A total of 15 out of 17 donors showed measurable IFNγ^+^ CD8^+^ T-cell responses against at least one of the previously reported HLA-A2 HPV16 epitopes. These 15 donors were categorized as HPV16^+^ (responsive). Interestingly, despite all being HLA-A2-positive and stimulated with HLA-A2-specific peptides, the different HPV16-reactive donors exhibited considerable variability in the antigen specificity of their elicited T-cell responses (Figure 1A). Overlapping peptide pools were also able to stimulate both CD8^+^ and CD4^+^ IFNγ^+^ responses in most of the donors, predominantly towards E1 and, to some extent, towards E7 (Figure 1B,C).

### 3.2. HPV16 Peptide-Primed PBMCs Can Specifically Recognize HPV16^+^ Ca Ski Cancer Cells

PBMCs from one of the HPV16-reactive donors, donor 16, were chosen for further evaluation to address whether the detected peptide-specific T cells could recognize HPV16^+^ cancer cells. The PBMCs were primed with HPV16 single peptides and cultured for two weeks. Then, the PBMCs were shortly co-cultured with Ca Ski cells, an HLA-A2^+^ HPV16^+^ cervical carcinoma cell line, and the CD8^+^ T-cell responses to Ca Ski cells were measured by ICS and flow cytometry. We found Ca Ski-reactive CD8^+^ T cells in all the PBMC cultures (Figure 2A). The PBMCs that had been cultured with peptide E2 aa138-147 were especially reactive towards the Ca Ski cells in this donor, which is also seen in Figure 1A. All peptide-reactive CD8^+^ T cells, except when primed with peptide E6 aa18-26 and peptide E6 aa29-38, secreted both TNFα and IFNγ cytokines (Figure 2B) and the CD107a CD8^+^ T-cell degranulation marker (Figure 2C), indicating high T-cell functionality.

### 3.3. Verification of the Human Ex Vivo Model of Immunization for Boosting of HPV16-Specific CD8^+^ T Cells

Our aim was to develop a new model for investigating vaccine-induced T cells in a human ex vivo setting. Nucleic-acids-based vaccines are front runners in current therapeutic vaccine development, and in this study, we based our assay development on adenoviral-vectored vaccines. Briefly, our suggested model is based on the transduction of DCs with the vaccine and the co-culture of these vaccine-transduced DCs with syngeneic PBMCs in order to expand vaccine-specific T cells. Adenoviral-vector-based vaccines are known to be strong inducers of T-cell responses, and encoding the MHC class II invariant chain (Ii) in viral vector vaccines together with the vaccine transgene has been reported to enhance the induction of CD4^+^ and CD8^+^ T-cell responses [34] and functionality of CD8^+^ T cells [31].

Initially, we assessed the boosting capacity of vaccine-transduced DCs in comparison with single peptide stimulation. PBMCs from donor 16 were primed with HLA-A2-restricted HPV16 single peptides derived from either HPV16 E1, E2, E6, or E7 protein and cultured for two weeks in the presence of a low dose of IL-2. On day 14, the primed T cells were then boosted with either the same single peptides used for the prime or by co-culturing with Ad19-transduced DCs encoding HPV16 with (Ad19-Ii-E1E2E6E7) or without (Ad19-E1E2E6E7) Ii or without any gene inserted (Ad19-NegCtrl, empty vector) to assess the background responses. One week after the boost, on day 21, the reactivity was evaluated by stimulation with the cognate HPV16 single peptides and intracellular flow cytometry (Figure 3A).

We saw that DCs transduced with the Ad19-HPV16 vaccines (Ad19-Ii-E1E2E6E7 and Ad19-E1E2E6E7) elicited higher frequencies of HPV16-specific IFNγ^+^ CD8^+^ T cells, compared to boosting with single HPV16 peptides, and that Ii (Ad19-Ii-E1E2E6E7) significantly increased this boosting effect (Figure 3B), reflecting the reality of in vivo immunizations [34]. Therefore, these results support that the co-culture of PBMCs with vaccine-transduced syngeneic DCs may constitute a feasible and potent ex vivo model of in vivo immunizations.

For the different experiments presented here, each required a cautious selection of the most suitable negative controls to use for an adequate subtraction of the background signal. For the peptide cultures presented in Figure 1 and Figure 2, the background signal was based on cells from the respective peptide cultures, which were not restimulated prior to the ICS. However, when using Adv-transduced DCs for priming or boosting, the scenario is a little more complex. Interestingly, when stimulating the PBMCs by co-culturing with vaccine-transduced DCs, we observed a slight increase in the proportion of CD8^+^ T cells (Appendix AA) and a higher level of CD8^+^ T-cell reactivity in the absence of stimulation prior to ICS compared to the single HPV16 peptide stimulation (Appendix AB). This enhanced signal is likely due to the continued expression of the Ad19-HPV16-encoded antigens upon DC transduction and prolonged epitope display on their cell surface (Appendix AC), providing longer-term T-cell stimulation. Therefore, an empty Ad19 vector encoding no antigens (Ad19-NegCtrl) was included as a background control for the vaccine-transduced DC co-cultured samples.

### 3.4. Verification of the Human Ex Vivo Immunization Model for the Priming of HPV16-Specific CD8^+^ T Cells

Next, we used the human ex vivo DC-transduction-based assay to simulate a primary in vivo Adv-vector-based immunization without prior peptide-priming of the T-cell response. The PBMCs of three HPV16-reactive donors (donors 9, 13, and 14) were primed with DCs transduced with Ad19-Ii-E1E2E6E7 or with a pool of peptides (pp) covering the entire HPV16 E1, E2, E6, and E7 proteins, to allow a direct comparison of delivery of a large amount of antigen (HPV16 E1, E2, E6, and E7 comprise 1272 amino acids) either by peptides or by adenoviral-based-vectors. One week after the prime, we restimulated with HPV16 E1, E2, E6, or E7 peptide pools (separately), and the reactivity was assessed by ICS and flow cytometry.

In the two responsive donors (donors 9 and 13), Ad19-Ii-E1E2E6E7-transduced DCs were superior at priming HPV16-specific CD8^+^ T-cell responses compared to the pool of E1, E2, E6, and E7 peptides (Figure 4A,C). Donor 14 had a very high background response after priming with Ad19-NegCtrl (empty vector)-transduced DCs, which resulted in a complete masking of any actual HPV16-specific responses. Notably, the peptide pool prime did not induce detectable IFNγ^+^ CD8^+^ HPV16 T-cell responses in any of the donors (Figure 4A–C).

This indicates that the human ex vivo immunization assay can efficiently prime HPV16-specific CD8^+^ T-cell responses, although the adenoviral-vector background signal, due to the stimulation of pre-existing adenovirus-specific T cells, may interfere with the readout in a prime-only setting.

### 3.5. Use of the Ex Vivo Human Immunization Model for the Comparison of Different Prime-Boost Strategies

Due to the induction of anti-vector antibodies, prime-boost immunization regimens employing viral vectors should be heterologous in order to achieve optimal T-cell responses. This is commonly achieved by using vectors of different viral origin [35,36], alternating adenoviral serotypes [21,22,23,25,26], or other vaccine technologies [37].

We employed the ex vivo immunization assay to compare different combinations of prime and boost modalities, where HPV16 E1, E2, E6, and E7 were delivered either as a pool of overlapping peptides or by DCs transduced by the different Adv serotypes Ad5, Ad19, or Ad5f35. Although the use of adenoviral-vector serotypes is restricted for clinical use, here we used different vector combinations to elucidate whether any booster regimen could immunologically outperform the others in order to generate T cells for ex vivo analysis.

One week after the boost, T-cell responses were measured by stimulation with HPV16 E1, E2, E6, or E7 peptide pools and an intracellular flow cytometry readout. The results showed a high degree of variation among the three donors included in the assay. In a highly responsive donor (donor 13, Figure 5A), all prime-boost regimens elicited strong IFNγ^+^ CD8^+^ responses, especially toward the E1 HPV16 peptide pool. In contrast, in an almost non-responsive donor (donor 14), the different prime-boost modalities showed no IFNγ secretion upon restimulation (Figure 5B), most likely due to a high level of background signal, as also observed in the previous prime-only experiments (Figure 4B and Appendix AB). Donor 9 showed an intermediate level of response with distinct differences in reactivity against the different HPV16 antigens (E1, E2, E6, and E7) depending on the prime-boost regimen. Notably, prime-boost with peptide pools did not elicit IFNγ^+^ CD8^+^ T-cell responses in donor 9; however, responses from all the other regimens could be obtained, which included stimulation by adenoviral vectors (Figure 5C). Overall, all the different adenoviral vaccination strategies successfully expanded HPV16-specific IFNγ-secreting CD8^+^ T cells in two out of three donors, with high background masking potential responses in the last donor.

### 3.6. Only Vaccine-Expanded/Sorted HPV16-E1-Specific T Cells Are Functional Regarding Killing of Cervical Carcinoma Cancer Cells

Given the high background observed in donor 14, we wished to investigate if this donor was, in fact, truly unresponsive or whether the lack of response was due to a combination of high background and a low fraction of initial HPV16-responding T cells and whether the model could be optimized to remove the masking background responses.

PBMCs of donor 14 (low responder) and donor 13 (high responder) were primed by Ad19-Ii-E1E2E6E7-transduced DCs (or Ad19-NegCtrl) and boosted one week later with Ad5- or Ad5f35-Ii-E1E2E6E7-transduced DCs (or Ad5-NegCtrl) similarly to Figure 5. One week after boosting, cells were stimulated with DCs transduced with either Ad5- or Ad5f35-Ii-E1E2E6E7 (heterologous; not matching the previous boost regimen), or a pool of HPV16 E1 peptides (E1pp) in order to further investigate the strong reactivity of E1 seen in Figure 1 and Figure 5. Further, 5 h after stimulation, activated T cells were magnetically sorted based on the IFNγ secretion, and the IFNγ-positive cells were further expanded by a rapid T-cell expansion protocol (REP) [33]. Fifteen days later, T cells were harvested and restimulated with each of the HPV16 E1, E2, E6, or E7 peptide pools, and the immune responses were assessed by ICS (Figure 6A).

Encouragingly, the IFNγ-based sorting and rapid expansion showed a substantial enrichment of HPV16-specific CD8^+^ T cells in donor 13 regardless of which regimen was used for the expansion and sorting. Especially for the cells expanded by the Ad19/Ad5f35-Ii-E1E2E6E7 prime-boost regimen, and sorted with E1 HPV16 peptide pool stimulation, the fraction of E1 HPV16-specific CD8^+^ T cells increased from less than 10% (Figure 5A, Ad19-Ad5f35) to making up almost 80% of the CD8^+^ T cells (Figure 6B, Ad19-Ad5f35 (E1pp)). For the previously unresponsive donor 14, the sorting and rapid expansion successfully revealed the presence of HPV16-specific CD8^+^ T cells, although at a much lower fraction of the total CD8^+^ T cells compared to donor 13 (Figure 6D).

In all cases, regardless of how the expansion and the sorting were carried out, CD8^+^ T-cell responses were only detected against HPV16 E1. While this was expected for the cells sorted by E1 reactivity, it was surprising to observe this level of E1-specific CD8^+^ T-cell dominance for the T cells sorted by their reactivity towards DCs transduced with adenoviral vectors containing all E1, E2, E6, and E7 HPV16 proteins (Figure 6B,C and Appendix AA,B). In addition to the HPV16-specific CD8^+^ T-cell responses, the CD4^+^ IFNγ^+^ (and TNFα^+^) T-cell responses directed primarily toward E1 were also observed (Figure 6D,E and Appendix AC,D). CD4^+^ responses against E7 were detected in a sample sorted by E1 recognition (Figure 6E and Appendix AD), probably due to high levels of unspecific background reactivity before the rapid expansion, as previously seen in Figure 5B.

The cancer-killing capacity of the vaccine-expanded T cells was evaluated by a chromium-release assay that involves co-culturing different dilutions of effector T cells with ^51^Cr-labeled HPV16^+^ cervical carcinoma cancer cells (Ca Ski) and detecting ^51^Cr release as a measure of T-cell cancer-specific cytotoxicity. In this experiment, we saw that the T cells sorted by vaccine-transduced DCs stimulation were superior at killing Ca Ski cells, compared to the ones sorted by E1 peptide pool stimulation (Figure 6F,G), despite the latter setup being the one that generally yielded the highest percentage of E1 HPV16-specific IFNγ-secreting T cells (Figure 6B–E).

Indeed, when donor 13 effector T cells were assessed for their reactivity towards Ca Ski cells, there was a complete correlation between Ca Ski reactivity (recognition) (Figure 6H) and killing activity (Figure 6F). Increased surface CD107a was observed on the CD8^+^ T cells of these cultures as a complimentary readout of their cytotoxic capacity (Figure 6I). The recognition of Ca Ski cells was mostly MHC-I-mediated, as seen by a reduced fraction of IFNγ-producing CD8^+^ T cells (Figure 6H, open vs. filled bars) and a complete absence of degranulation (Figure 6I) when donor 13’s effector T cells were co-cultured with Ca Ski cells in the presence of an HLA-A2-blocking antibody. Notably, the dominance of E1-specific reactivity was maintained over time (Appendix AA,B). It is worth mentioning that the fraction of reactive cells out of the CD8^+^ T-cell population in donor 13 was slightly low in this assay (Appendix AA) compared to the previously observed fractions (Figure 6B), possibly due to the freezing and thawing processes (Figure 6A).

## 4. Discussion

Here, we evaluate a new human ex vivo PBMC assay that is designed to mimic certain aspects of a human in vivo immunization with Adv-vector-based vaccines and use it to evaluate the immunogenicity, over time immunodominance, and anti-tumor ability of a novel therapeutic HPV vaccine candidate encoding E1E2E6E7 HPV16 genes fused to the T-cell adjuvant Ii.

We show that this human ex vivo immunization model can be effectively used to restimulate and expand reactive and functional T-cell responses specific to the vaccine antigens. The magnitude of the response elicited by Adv vectors was superior to what could be obtained by simple peptide stimulation. We observed high levels of background responses, which are probably due to the high reactivity to the vaccine adenoviral vector itself, and in some cases, this drowned the early antigen-specific responses (Figure 4 and Figure 5, donor 14). However, this obstacle of detecting low-magnitude antigen-specific T-cell responses could be overcome by employing IFNγ-sorting and rapid expansion steps (Figure 6, donor 14).

The current preclinical models for the evaluation of cancer immunotherapies constitute a translational constraint, also for HPV-associated disease. The human ex vivo PBMC assay presented here has the advantage of being a fully human method for the stimulation of antigen-specific T cells, enabling expansion to sufficient numbers and allowing for further testing. Some of the constraints of the assay are the fact that ex vivo assays do not comprehensively represent a live animal, as well as a notable degree of inter-donor variability (although this also reflects the reality of human diversity), and the fact that the sorting step needed to improve the signal:noise ratio, making the assay less suitable for quantitative comparisons. However, the assay can reveal whether specific targets are immunogenic or not and provide qualitative data on the induced responses, especially regarding immunodominant determinants.

It is highly relevant to further validate the ex vivo human PBMC assay for specific T-cell expansion in a larger number of donors with diverse HLA types, including the HLA alleles reported to be relevant for clinical outcomes, such as HLA-DQB-2 [38], as well as to evaluate the functionality and relevance of this assay for therapeutic vaccination and its feasibility for other targets. Importantly, this ex vivo assay may be used on blood samples from patients and can thereby provide valuable insights into the immunogenic capabilities of a therapeutic vaccine candidate in a target patient population. Such information could provide a valuable tool for improving the selection of therapeutic vaccine candidates with the best possible chance of helping patients and of identifying patients who have the most benefit from treatments that rely partly or fully on the cancer-specific T-cell response.

The evaluation of the HPV therapeutic vaccine showed that E1 was the most immunogenic of the four HPV genes included in the vaccine. Furthermore, the data suggests that although T cells from HPV16-reactive donors initially react to a palette of HPV16 antigens, the ones that persist over time are those specific for E1. Whether E1 is just more immunogenic than the other HPV16 antigens, possibly by being longer and thereby providing a greater number of potential T-cell epitopes, or whether T cells recognizing non-E1 antigens are more prone to exhaustion during the culture, remains unknown. Despite this, T-cell exhaustion caused by continuous exposure to antigens is a common feature of HPV [39,40], and it has been shown that the exhausted T-cell phenotype was specifically correlated with E7 expression in patients with HPV^+^ HNSCC [40]. We have previously hypothesized that E1 is a relevant target that is poorly immunogenic in patients and, therefore, does not commonly induce exhaustion [13]. This could explain why we see that E1-specific CD8^+^ T cells persisted over time upon repeated stimulation through the human ex vivo immunization assay.

The finding of E1 immunodominance is in line with previous published murine studies using the same vaccine design [21,31]. Currently, most therapeutic HPV vaccines target E6 and E7 only, but our data suggest E1 as an immunogenic and promising antigen as well, and other recently published studies also argue for the potential of targeting HPV antigens beyond E6 and E7. A study by Eberhardt et al. on patients with HPV-positive head and neck cancer showed that more patients had tumor-infiltrating CD8^+^ T cells towards E2 rather than E6 or E7 [18]. McInnis et al. showed strong anti-E1 and E2 CD8^+^ T-cell responses in head–neck cancer patients and also reported that E1 was expressed in tumor biopsies from all patients enrolled in their study [20]. Similarly, Peng et al. reported an E1 expression in all enrolled cervical cancer patients. Together, this indicates the advantages of the inclusion of E1 and E2 in addition to E6 and E7 in therapeutic HPV vaccine designs [19]. Notably, it has been shown in cervical cancer patients that detection, even of low frequencies of E1-specific T cells, correlated with a dramatic improvement in progression-free survival [41]. Similarly, E2-specific T cells have been shown to correlate with the absence of progression in women with cervical dysplasia [42,43].

We observed in Figure 6B–G that cells sorted by E1 peptide pool reactivity were highly reactive towards the E1 peptide pool 15 days later; however, this response correlated poorly with the recognition and killing of HPV16^+^ cancer cells. The opposite was seen for the samples sorted by reactivity towards DCs transduced by Advs encoding HPV16 E1, E2, E6, and E7, where the reactivity towards HPV16 peptide pools was relatively lower but where recognition and killing of HPV16^+^ cancer cells was evident. One explanation for this discrepancy could be that E1-specific T cells are not sufficient for an anti-cancer effect alone and that some undetectable T cells responding towards the other HPV antigens are responsible for the enhanced killing. Another explanation could be that even though the E1 peptide pool stimulation for sorting yielded the biggest breadth of E1-specific T-cell responses, it might not have stimulated the T-cell specificities that could specifically recognize and kill Ca Ski cells. Already trimmed peptide pools can result in a limited selection of peptides presented on MHC molecules in comparison to Adv-mediated antigen delivery [44]. The constraints of using peptide pools for stimulation were also reflected in the discrepancies observed between donors in Figure 1A,C, where a donor (e.g., donor 3) exhibits a robust CD8 response when stimulated with an HLA-A2-restricted HPV16 single peptide (e.g., E1 aa253-262) but then shows no response when stimulated with the corresponding peptide pool (e.g., E1 peptide pool). These results highlight the potential of using nucleotide-based approaches (DNA, RNA, and viral vectors) encoding full-length viral genes to generate potent and specific cellular responses capable of eradicating tumor cells and their potential application for T-cell therapy to treat HPV-related diseases.

Of note is that while T-cell responses towards all peptide pools (E1, E2, E6, or E7) were detected in Figure 1B,C, no peptide pool-specific T cells could be detected in Figure 4 (legend: pp) when the peptide pools were combined for stimulation intending to mimic the design of our Adv-based vaccine. While keeping in mind the risk of interassay variation, this difference could be due to peptide overload and/or immune competition. In contrast, Adv-transduced DCs could raise T-cell responses towards the HPV16 antigens, as shown in Figure 4 (legend: Ad19), suggesting a qualitative difference between the use of peptide pools and Adv for the delivery of large amounts of antigens to APCs. Additionally, it can be speculated that the use of Ii as an adjuvant also plays a role in this enhanced T-cell activation and broad T-cell specificity (Figure 3B) (legend: Ad19-Ii-E1E2E6E7), as Ii has previously been shown to both enhance the magnitude and the antigenic breadth of the T-cell response [34,45,46,47].

## 5. Conclusions

The data presented in this work indicates that the human ex vivo immunization assay can be a valuable tool for guiding future therapeutic vaccine development in combination with the other existing preclinical models. This assay not only serves as a powerful tool to increase the initial amount of vaccine antigen-specific T cells to allow further testing but also represents an indirect human model that mimics certain aspects of human immunization ex vivo, allowing to elucidate the vaccine immunogenicity, over time immunohierarchy, and cytotoxic capacity of the vaccine-expanded T cells.

Furthermore, this work supports the idea of using adenoviral-vector-based vaccines to deliver more and longer full-length antigens, such as the E1 HPV gene fused to the Ii T-cell adjuvant in the antigen design of HPV therapeutic vaccines. This would increase the chances of eliciting clinically relevant, non-exhausted HPV-specific T-cell responses to fight HPV infection and HPV-associated cancers.

## Figures and Tables

**Figure 1 cancers-15-05863-f001:**
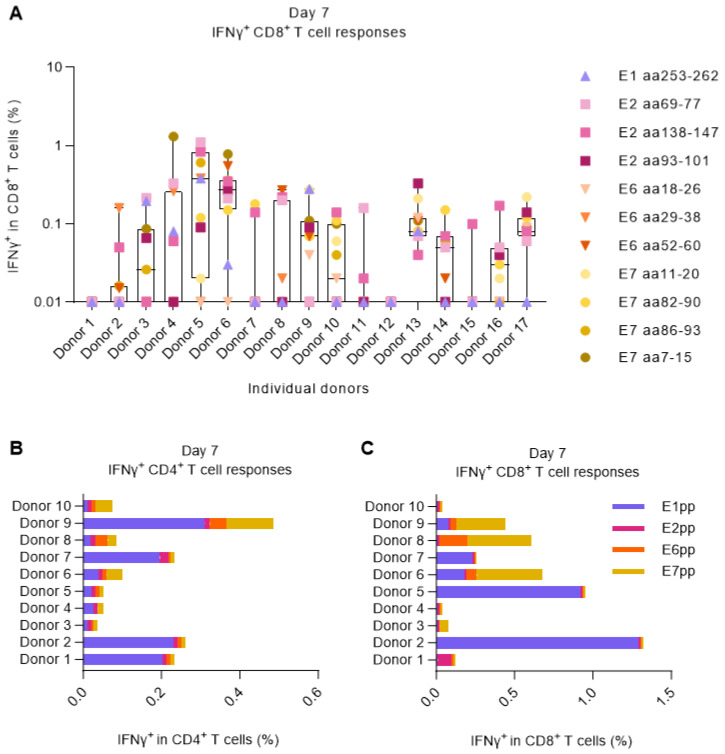
Screening for HPV16^+^ donors. PBMCs from 17 HLA-A2^+^ healthy donors were left unstimulated or were primed with 11 different HLA-A2-restricted HPV16 single peptides or HPV16 E1, E2, E6, and E7 peptide pools. After 7 days, PBMCs were either left unstimulated to be used as negative controls (background) or were restimulated with the matching peptides/pools. Data were analyzed by flow cytometry following the gating strategy illustrated in Appendix A. (**A**) Fraction (%) of IFNγ^+^ in CD8^+^ T cells upon single peptide stimulation showed that almost 90% of the donors responded to at least one of the peptides. The data shown here are a combination of two separate experiments performed under the same experimental conditions. Left Y-axis was set at the Log 10 scale, and thus, zero or minus values were converted to 0.01. (**B**,**C**) Fraction (%) of IFNγ^+^ in CD4^+^ and CD8^+^ T cells upon E1, E2, E6, or E7 HPV16 peptide pool stimulation shows that most donors could respond to peptide pool stimulation, especially to E1.

**Figure 2 cancers-15-05863-f002:**
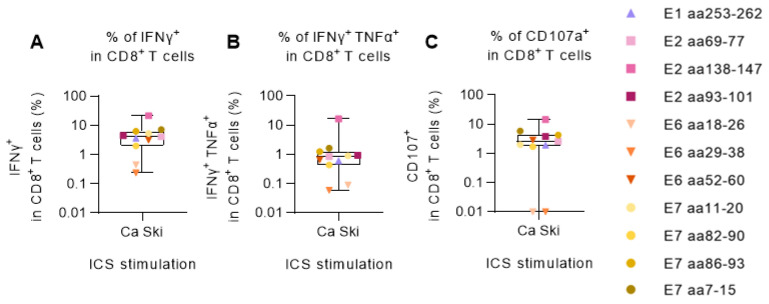
Degranulation and IFNγ and TNFα production of HPV16 peptide-expanded CD8^+^ T cells upon Ca Ski stimulation. PBMCs from the HLA-A2^+^ HPV16-reactive donor 16 were primed with single HLA-A2-restricted HPV16 peptides or were left unstimulated and cultured for two weeks. Each stimulation condition was co-cultured at a 5:1 ratio with Ca Ski cancer cells. After 5 h, cells were surface stained for CD107a and intracellularly stained for IFNγ and TNFα cytokines. Unstimulated (unprimed) samples co-cultured with Ca Ski cells were used as negative controls (background). Flow cytometry data were analyzed following the gating strategy illustrated in Appendix A. Left Y-axis was set at the Log 10 scale, and thus, zero or minus values were converted to 0.01. (**A**) Fraction (%) of IFNγ^+^ in CD8^+^ T cells upon Ca Ski stimulation showed that all peptide-expanded CD8^+^ T cells were able to recognize and become activated upon encountering the target cancer cells. (**B**) Fraction (%) of double-positive (IFNγ^+^ TNFα^+^) in CD8^+^ T cells upon Ca Ski stimulation showing the quality of the CD8^+^ T-cell responses. (**C**) Fraction (%) of CD107a^+^ in CD8^+^ T cells upon Ca Ski stimulation illustrated the capacity of the peptide-expanded cytotoxic T cells, excluding peptide E6 aa18-26 and peptide E6 aa29-38, to degranulate upon encountering the target cancer cells.

**Figure 3 cancers-15-05863-f003:**
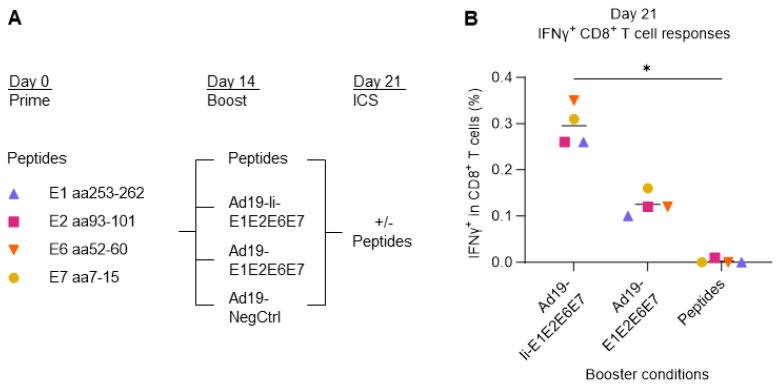
Comparison of different boosting conditions upon peptide T-cell prime and expansion. (**A**) Schematic representation of the experiment design. PBMCs from donor 16 were primed with either E1 aa253-262, E2 aa93-101, E6 aa52-60, or E7 aa7-15 HLA-A2-restricted single HPV16 peptides (peptides). Two weeks later, they were boosted with either cognate peptides or co-cultured at a 5:1 ratio with Ad19-transduced DCs encoding HPV16 with (Ad19-Ii-E1E2E6E7) or without (Ad19-Ii-E1E2E6E7) Ii. Ad19-NegCtrl (empty vector)-transduced DCs were used as negative controls for the Ad19-HPV16-boosted T cells. One week after the boost, the expanded T cells were either left unstimulated, or they were restimulated with either E1 aa253-262, E2 aa93-101, E6 aa52-60, or E7 aa7-15 HLA-A2-restricted single HPV16 peptides (peptides) for CD8 IFNγ staining. Flow cytometry data were analyzed following the gating strategy illustrated in Appendix A. Peptide homologous prime-boosted samples left unstimulated for the ICS were used as background and were subtracted from peptide homologous prime-boosted ICS-stimulated samples. Peptide-primed Ad19-NegCtrl-boosted samples, left unstimulated for ICS, were used to measure background responses and were subtracted from peptide-primed, Ad19-HPV16-boosted ICS-stimulated samples. (**B**) Fraction (%) of IFNγ^+^ in CD8^+^ T cells reacting to HPV16 peptides. Ad19-Ii-E1E2E6E7-transduced DCs showed to be the most immunogenic and, thus, better at boosting IFNγ^+^ CD8^+^ T-cell responses than Ad19-E1E2E6E7-transduced DCs and HPV16 single peptides. * *p* < 0.05: One-way ANOVA (non-parametric test), comparing the mean rank of each column with the mean rank of every other column.

**Figure 4 cancers-15-05863-f004:**
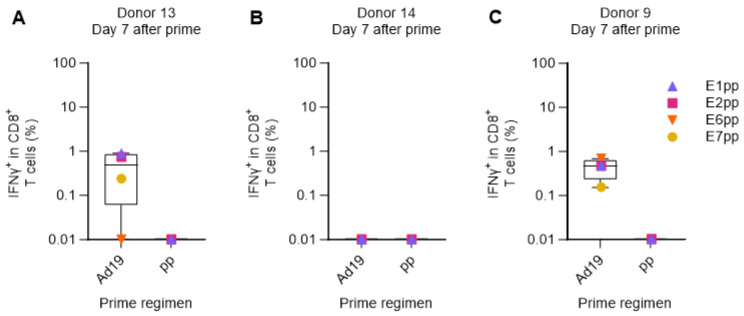
Comparison of Adv-transduced DCs and peptide pools for priming HPV16-specific T cells. PBMCs from donors 9, 13, and 14 were primed with either HPV16 E1, E2, E6, and E7 peptide pools (pp), Ad19-Ii-E1E2E6E7 HPV16 (Ad19), or Ad19-NegCtrl (empty vector)-transduced DCs at a 1:10 ratio. One week after, PBMCs were left unstimulated or were restimulated with HPV16 E1, E2, E6, or E7 peptide pools. CD8^+^ T cells were intracellularly stained for IFNγ and analyzed by flow cytometry. Ad19-NegCtrl-unstimulated samples were used as background and were subtracted from Ad19-Ii-E1E2E6E7-stimulated samples. HPV16 peptide-pool-primed unstimulated samples were used as background and were subtracted from HPV16 peptide-pool-primed stimulated samples. (**A**–**C**) Percentage (%) of IFNγ^+^ in CD8^+^ cytotoxic T lymphocytes 7 days after priming from donors 13, 14 and 9, respectively. Flow cytometry data were analyzed following the gating strategy illustrated in Appendix A. Left Y-axis was set at the Log 10 scale, and thus, zero or minus values were converted to 0.01.

**Figure 5 cancers-15-05863-f005:**
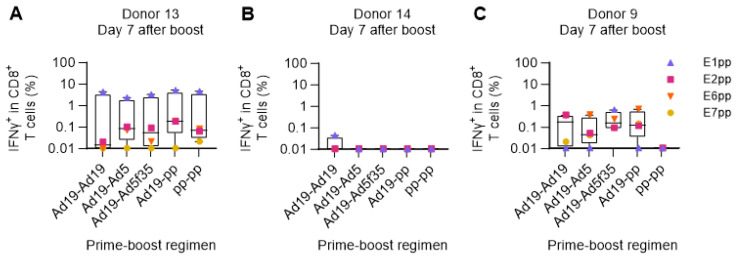
Comparison of different boosting strategies. PBMCs from donors 13, 14, and 9 were primed with either Ad19-Ii-E1E2E6E7 HPV16-transduced DCs (Ad19) or the combination of all HPV16 E1, E2, E6, and E7 peptide pools (pp). After 7 days, HPV16 peptide-pool-primed PBMCs were boosted with the same peptide pools (pp-pp), while Ad19-HPV16-transduced DC-primed PBMCs were boosted with either HPV16 peptide pools (Ad19-pp) or Ad19- (Ad19-Ad19), Ad5- (Ad19-Ad5), or Ad5f35- (Ad19-Ad5f35) Ii-E1E2E6E7-transduced DCs. Seven days after the boost, T cells were restimulated with HPV16 individual peptide pools, and CD8^+^ T cells were intracellularly stained for IFNγ production and analyzed by flow cytometry. Flow cytometry data were analyzed following the gating strategy illustrated in Appendix A. (**A**–**C**) Fraction (%) of IFNγ^+^ in CD8^+^ T cells from donors 13, 14, and 9, respectively, reacting to the different HPV16 peptide pools after different prime-boost stimulation regimens. Prime-boost T cells left unstimulated during ICS were used as background and were subtracted from the ICS HPV16 peptide-pool-stimulated samples. Donor 14’s Ad19-Ad19-unstimulated sample was lost during the flow cytometry run, and thus, the mean of donor 9’s and donor 13’s Ad19-Ad19-unstimulated samples were used as background for donor 14’s Ad19-Ad19-stimulated sample, which would explain why it is the only condition in which we observed a minimal response. Left Y-axis was set at the Log 10 scale, and thus, zero or minus values were converted to 0.01.

**Figure 6 cancers-15-05863-f006:**
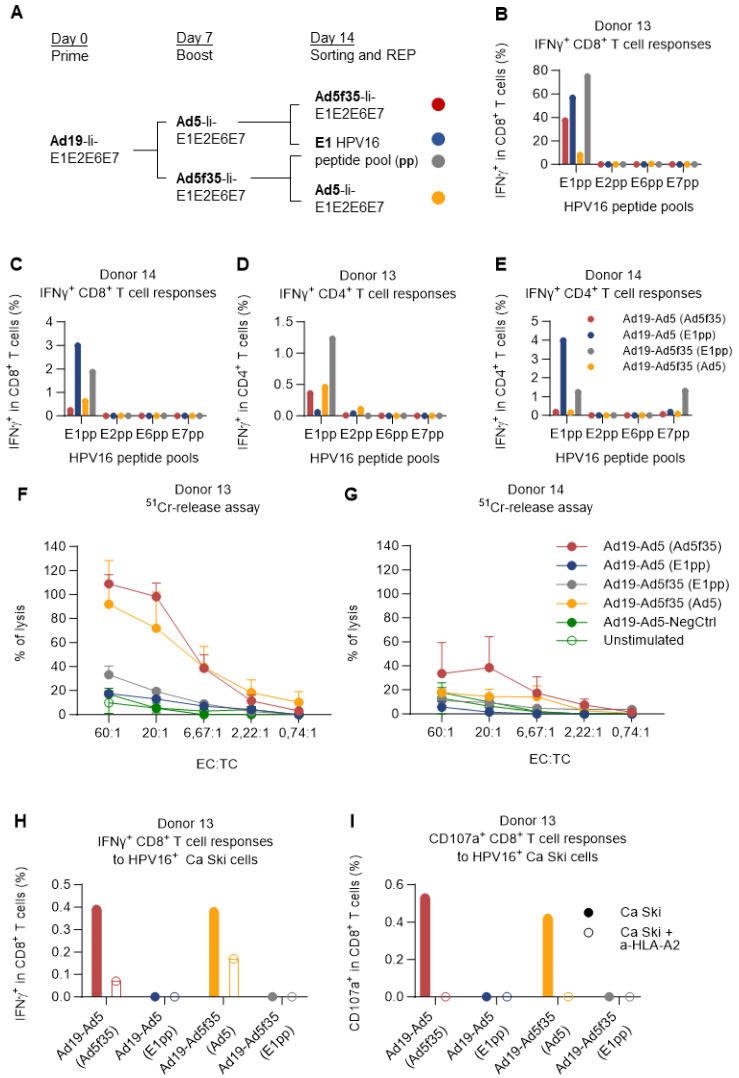
E1 T-cell specificity dominated over time but only induced the killing of Ca Ski cells when delivered via an Adv vector. (**A**) Schematic representation of the experiment timeline. PBMCs from donor 13 and donor 14 were prime-boosted with Ad19-Ad5/Ad5f35-Ii-E1E2E6E7 HPV16-transduced DCs and were further stimulated with either the HPV16 E1 peptide pool or with Ad5- or Ad5f35-Ii-E1E2E6E7 HPV16-transduced DCs for 5 h. Activated IFNγ-producing T cells were magnetically sorted into REP cultures. Fifteen days after REP, T-cell reactivity against the different HPV16 peptide pools was tested by flow cytometry, and their cytolytic activity was assessed by ^51^Cr-release assay against Ca Ski cancer cells. The remaining T cells were frozen, and donor 13’s expanded T cells were thawed to reassess their antigen specificity and their capacity to secrete IFNγ and degranulate upon Ca Ski recognition by flow cytometry. T cells left unstimulated during ICS were used as background and were subtracted from the peptide pool-stimulated samples. (**B**,**C**) Fraction (%) of IFNγ^+^ T cells in CD8^+^ T cells from donor 13 and donor 14 showing only E1^+^ IFNγ^+^ CD8^+^ T-cell responses for all prime-boost and sorting conditions. (**D**,**E**) Fraction (%) of IFNγ^+^ in CD4^+^ T cells from donor 13 and donor 14 showing mainly E1^+^ IFNγ-secreting CD8^+^ T cells for all prime-boost and sorting conditions, except for the Ad19-Ad5f35 prime-boosted T cells sorted by E1pp reactivity, which also showed some reactivity towards the E7pp. (**F**,**G**) Use of the ^51^Cr release assay to determine the cytotoxic activity of different expanded and sorted effector T cells after 4 h of incubation with Ca Ski cancer cells. T cells stimulated with Ad19- and Ad5-NegCtrl (empty vectors)-transduced DCs (filled green circles) and unstimulated PBMCs (empty green circles) from each of the donors were used as negative controls for this assay. EC:TC ratios indicate the ratio between effector T cells and target (Ca Ski) cells. Percentages of lysis above the negative controls (green lines) are considered specific. (**F**) Percentage (%) of lysis of different effector T cells from donor 13, showing that only effector T cells sorted using Adv-Ii-E1E2E6E7-transduced DCs possessed cytolytic activity and were capable of specifically lysing Ca Ski cancer cells. (**G**) Percentage (%) of lysis of different effector T cells from donor 14, showing that mainly effector T cells sorted using Ad5f35-Ii-E1E2E6E7-transduced DCs (red line) and, to a lesser extent (and only for some EC:TC ratios), effector T cells sorted using Ad5-Ii-E1E2E6E7-transduced DCs (yellow line), were capable of specifically lysing Ca Ski cancer cells. (**H**,**I**) Donor 13’s frozen effector T cells were thawed to confirm the maintenance of antigen specificity over time and their capacity to recognize Ca Ski cells. Fraction (%) of IFNγ^+^ CD8^+^ T cells responding to HPV16 peptide pools showed retention of E1 immunodominance. Only Adv-transduced sorted DCs showed specific recognition (IFNγ secretion) and degranulation (CD107a) upon co-culture with Ca Ski cells in an HLA-A2^+^-restricted fashion. Flow cytometry data were analyzed following the gating strategy illustrated in Appendix A.

## Data Availability

The data presented in this study are available on request from the corresponding author. The data are not publicly available due to privacy restrictions.

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
