# Peer review of "Preferential Expansion of HPV16 E1-Specific T Cells from Healthy Donors’ PBMCs after Ex Vivo Immunization with an E1E2E6E7 Fusion Antigen"

_cancers, 2023, doi:10.3390/cancers15245863_

Round 1

Reviewer 1 Report

Comments and Suggestions for Authors

The author established a new experimental model that simulates the human immune system by co-cultivating dendritic cells (DC) with peripheral blood mononuclear cells (PBMCs). This model was used to evaluate a therapeutic vaccine using an adenoviral vector. This new method allowed us to demonstrate the functional effects of vaccine-transduced DC on T cells and revealed immune hierarchy information showing the long-term advantage of E1-specific T cell immunity over time. The research objectives of this study have certain implications for the development of therapeutic vaccines for cervical cancer. The logical flow of the study is coherent, the experimental figures are uniformly styled, and the formatting is relatively smooth. However, there are some suggestions and concerns regarding certain details and experiments. It is recommended that the author address and respond to the following:

1.  We suggest combining the collection and cryopreservation of PBMCs under one heading in the Materials and Methods section.

2.  Will the widespread use of adenovirus Ad5 induce a population immune response against Ad5? Is this vaccine suitable for evaluating the assessment model in article?

3.  Are there negative controls of non-caski cell lines or more validation data of HPV16-transformed cell lines such as Siha in Figure 2?

4.  Please add the negative control for Figure 3B,additionally,the legend need to be adjusted as they could lead to ambiguity. Can the results showing an enhancement effect of 0% for peptides in the last group be confirmed?

5.  In Figure 5, there are some differences between different donors. Can this model be used as a universally applicable model?

6.  In line 112 of the introduction, the author states that "HPV16 E1-specific CD8+ T cells are reliably inducible and dominate long-term in non-biased T cell expansion cultures." Please confirm if a 14-day detection of T cell response can be considered long-term.

Comments on the Quality of English Language

The English writing in this article is relatively smooth and produces few ambiguities. It may require the editor to carefully check for language errors and the use of colloquial expressions.

Reviewer 2 Report

Comments and Suggestions for Authors

1-     The abstract, the introduction, the results and discussion were well presented and described.

2-     Why human adenovirus 5 which in its wild type cause mild to acute respiratory infection or human Adenovirus 19a/64   which in its wild type cause epidemic keratoconjunctivitis and their antibodies may neutralize these vectors and reduce their efficacy during treatment? Therefore, animal Adenovirus vectors could have a better outcome? I think should be added as a limitation of study in the discussion section.

3-     In addition to HLA-A2, why other HLA alleles such as HLA-DQB1 Restrict HPV16 E7 OR HLA-A HPV E6 were not used in this study to obtain better evaluation? (Ref.A,B)

 References:

A= Peng S, Trimble C, Wu L, Pardoll D, Roden R, Hung CF, Wu TC. HLA-DQB1* 02–restricted HPV-16 E7 peptide–specific CD4+ T-cell immune responses correlate with regression of HPV-16–associated high-grade squamous intraepithelial lesions. Clinical cancer research. 2007 Apr 15;13(8):2479-

B- Cai, H., Feng, Y., Fan, P. et al. HPV16 E6-specific T cell response and HLA-A alleles are related to the prognosis of patients with cervical cancer. Infect Agents Cancer 16, 61 (2021). https://doi.org/10.1186/s13027-021-00395-y

Reviewer 3 Report

Comments and Suggestions for Authors

The manuscript by Daradoumis et al. develops a model of immunization by using an E1E256E7 fusion antigen.

I think that this manuscript is very well written and the results are clear. This manuscript describes a strategy to use fusion antigens to develop vaccines against HPV16.

I think this manuscript is well-suited for the issue and it should be published. I do not detect any further issues to be discussed.

Reviewer 4 Report

Comments and Suggestions for Authors

Daradoumis et al. describe a new human ex vivo PBMC assay to study T-cell responses induced by an adenoviral therapeutic HPV vaccine. The findings are interesting, supporting other studies that other proteins then HPV16 E6 and E7 should be considered for therapeutic HPV vaccines.

The authors should address an apparent discrepancy between Fig. 1A and Fig.1C. For example, Donor 3 has a robust CD8 response when stimulated with the single peptide E1 aa253-262, but shows no response when stimulated with the E1 peptide pool. Or Donor 4, who shows CD8 responses upon stimulation with various single peptides, but hardly any response when stimulated with the different peptide pools.

The experiment with different adenoviral and peptide pool combinations, for which the results are shown in Figure 5, seems overly complicated. The authors mention that heterologous prime-boosting is expected to result in stronger immune responses. However, in the assay described here, with ex vivo stimulation, pre-existing antibodies against the virus do not play a role. Showing only one prime-boost combination would have been sufficient.

The authors mention the stimulation of E1 responses in the prime-boost regimens (Fig. 5), but do not mention the apparent suppression of E2 response in the Ad19-Ad19 combination or the suppression of E7 responses in all combinations with Ad19 as prime immunization. Do they have an explanation for these findings?

Minor comment:

Fig. 1 - It would be better if different colors were used for the different peptides per antigen. Now it is not possible, for example, the responses induced by the 3 different E2 peptides.

Fig. 5 - The symbols indicating the different peptide pool in Figure 5C should be moved to the right, now they seem to represent the responses in the 'pp-pp' column.

Round 2

Reviewer 1 Report

Comments and Suggestions for Authors

Thanks for the author‘s modification and answer to the above suggestions and questions. The author’s reply can solve the above problems. The whole work has innovative points, the research content is logical, and the results are credible.In my opinion,the paper could be published.